# Learning in continuous action space for developing high dimensional potential energy models

Sukriti Manna[1,2], Troy D. Loeffler[1,2], Rohit Batra[1], Suvo Banik[1,2], Henry Chan [1], Bilvin Varughese[1,2], Kiran Sasikumar[1], Michael Sternberg[1], Tom Peterka[3], Mathew J. Cherukara [4], Stephen K. Gray [1], Bobby G. Sumpter [5] & Subramanian K. R. S. Sankaranarayanan [1,2 ✉]

Reinforcement learning (RL) approaches that combine a tree search with deep learning have found remarkable success in searching exorbitantly large, albeit discrete action spaces, as in chess, Shogi and Go. Many real-world materials discovery and design applications, however, involve multi-dimensional search problems and learning domains that have continuous action spaces. Exploring high-dimensional potential energy models of materials is an example. Traditionally, these searches are time consuming (often several years for a single bulk system) and driven by human intuition and/or expertise and more recently by global/local optimization searches that have issues with convergence and/or do not scale well with the search dimensionality. Here, in a departure from discrete action and other gradient-based approaches, we introduce a RL strategy based on decision trees that incorporates modified rewards for improved exploration, efficient sampling during playouts and a "window scaling scheme" for enhanced exploitation, to enable efficient and scalable search for continuous action space problems. Using high-dimensional artificial landscapes and control RL problems, we successfully benchmark our approach against popular global optimization schemes and state of the art policy gradient methods, respectively. We demonstrate its efficacy to parameterize potential models (physics based and high-dimensional neural networks) for 54 different elemental systems across the periodic table as well as alloys. We analyze error trends across different elements in the latent space and trace their origin to elemental structural diversity and the smoothness of the element energy surface. Broadly, our RL strategy will be applicable to many other physical science problems involving search over continuous action spaces.

[1] Center for Nanoscale Materials, Argonne National Laboratory, Lemont, IL 60439, USA. [2] Department of Mechanical and Industrial Engineering, University of Illinois, Chicago, IL 60607, USA. [3] Math and Computer Science Division, Argonne National Laboratory, Lemont, IL 60439, USA. [4] Advanced Photon Source, Argonne National Laboratory, Lemont, IL 60439, USA. [5] Center for Nanophase Materials Sciences, Oak Ridge National Laboratory, Oak Ridge, TN 37831, USA. ✉email: skrssank@uic.edu

Reinforcement learning (RL) and decision tree (e.g., Monte Carlo tree search) based RL algorithms are emerging as powerful machine learning approaches, allowing a model to directly interact with and learn from the environment[1]. RL has achieved impressive capabilities with tremendous success in solving problems with intractable search space, for example in game playing (such as chess, Shogi, and Go)[2,3], chemical synthesis planning[4,5] or drug discovery[6]. However, these methods have been limited to discrete action space—e.g., "move pawn to e4", "add acetone reagent" or "remove chemical group -COOH"[7]. Many real-world problems including several grand challenges in materials discovery and design involve decision making and search via continuous action space[8]. These include, for instance, the problem of searching optimal model parameters/weights, exploring low-energy material phases or inverse design, optimizing experimental parameters, or synthesizing material properties[5,9]. While it is highly desirable to translate the merits of RL methods to solve search problems in materials design, a challenge lies in its continuous, complex and multidimensional nature and is further complicated by an exorbitantly large number of degenerate and/or suboptimal solutions.

One of the more successful versions of RL involves the use of Monte Carlo tree search (MCTS)[10,11], which utilizes playouts to select the best possible action (with maximum reward) from the current state (Fig. 1). Here, playouts refer to random actions from the model that allow it to learn by interacting with the environment. The larger the number of playouts, the better the model estimate of reward and the more promising is the model action selection. Notably, the MCTS performs this search in a tree structure, continuously growing either those leaves of the tree that result in a maximum reward (exploitation) or those which have not been adequately sampled (exploration)[12]. It is important to understand that when the action space is discrete, a parent leaf will exhibit limited possible child leaves, all (or some) of which can be assessed for their prominence. When the action space is continuous, the number of possible child leaves are infinite, irrespective of the depth of the parent leaf, making the use of MCTS seemingly impossible in continuous action space.

Recently, attempts to develop MCTS for continuous action space problems is gaining momentum[11,13]. A detailed discussion of the continuous action MCTS methods developed in this work is presented in the Supplementary Methods (Section 3.3). In a significant departure from traditional discrete MCTS[14] and recent continuous action space MCTS approaches[11,13], we introduce three concepts to tackle continuous action space problems: (1) a uniqueness function to avoid degeneracy, (2) correlating tree-depth to action space, and (3) implementing an adaptive sampling of playouts. The first ensures that only unique leaves are being explored during MCTS. This avoids the common issue of convergence of two initially separate MCTS branches to the same region of the continuous search space. More importantly, this resolves the problem of multiple representations of the same (degenerate) solution as often encountered in several physical problems (for example, a phase structure can be represented using varying unit cell definitions). The second concept provides a meaningful structure to the algorithm, with the child leaf searching within a narrower region of that of the parent node. Lastly, to improve the quality of the playouts, especially in the case of high-dimensional search space, the random simulations were biased to sample those regions that were closer to the parent leaf.

We deploy our approach to a representative high-dimensional and continuous parameter search for physics-based and neural network models that involves navigating through high-dimensional potential energy surfaces (PES)[15] of elemental nanoclusters and bulk systems. Historically, this has represented a major challenge for molecular modeling and has been accomplished using human intuition and expertise, requiring years of painstaking effort. Recently, a variety of global/local optimization methods have emerged for this task[16], but they either have convergence issues, do not scale well with the search dimensionality, or cannot incorporate important gradient-free knowledge (e.g., dynamic stability). Over several decades, these approaches have been used to develop a large number of multi-parameter physics-based models, mainly for bulk systems and their static/dynamical properties. The configurational diversity and the complex PES for nanoscale clusters, especially those far-from-equilibrium, pose a significant challenge. The extrapolation to capture nanoscale properties and dynamics, therefore, shows strong deviation from the ground truth (estimated using high-fidelity first-principles models, such as density functional theory). Here, we demonstrate the efficacy of our continuous action MCTS (c-MCTS) by developing a hybrid bond-order potential (an 18-dimensional parameter space) for 54 elements chosen across the periodic table, capturing a variety of bonding environments and demonstrating the generality, efficiency, and robustness of our approach. For each element, we train by fitting energies of thousands of carefully sampled (see Methods) nanoclusters of varying sizes, which are particularly known for their complex chemistry[17,18] and are difficult to train using traditional optimization strategies. Our ML-trained bond-order potentials show significant performance improvement over current physics-based potential models in terms of energies, atomic forces, and dynamic stability, and generalize well for dynamical properties that were not included during model training.

## Results

**Learning in continuous action space.** MCTS is a powerful algorithm for planning, optimization, and learning tasks owing to its generality, simplicity, low computational requirements, and a theoretical bound on exploration vs exploitation trade-off[10,14,19]. As illustrated in Fig. 1b, it utilizes a tree structure for the parameter search consisting of four key stages: (1) selection: based on a tree policy select the leaf node that has the highest current score; (2) expansion: add a child node to the selected leaf after taking a possible (unexplored) action; (3) simulation: from the selected node, perform Monte Carlo trials of possible actions using a playout policy to estimate the associated expected reward; (4) back-propagation: pass the rewards generated by the simulated episodes to update the scores of all the parent leaves encountered while moving up the tree. A popular tree policy to use is the upper confidence bound for parameters (UCP)[14,20]:

$$\text{UCP}(\theta_j) = -\min(r_1, r_2, ..., r_{n_i}) + c \cdot f(\theta_j) \cdot \sqrt{\frac{\ln N_i}{n_i}} \qquad (1)$$

where $\theta_j$ represents the node $j$ in the MCTS structure, $r$ denotes the reward for a given playout, $c(>0)$ is the exploration constant, $n_i$ is the number of playout samples taken by node $\theta_j$ and all of its child nodes, and $N_i$ is the same value as $n_i$ except for the parent node of $\theta_j$. ($f(\theta_j)$ is the uniqueness function specifically introduced in this work and is equal to 1 in traditional MCTS settings.) This policy tries to balance the search between those nodes which have either returned the maximum reward (left term) or have not been explored enough (right term). In contrast, the playout policy selects random actions (from a node) until the simulated episode is over.

Several issues deter the use of the traditional discrete as well as recently developed MCTS formulations for continuous action space. First, from any parent node, the number of possible child nodes is infinite as the action space is continuous. Further, for problems involving parameter search or optimization, the same

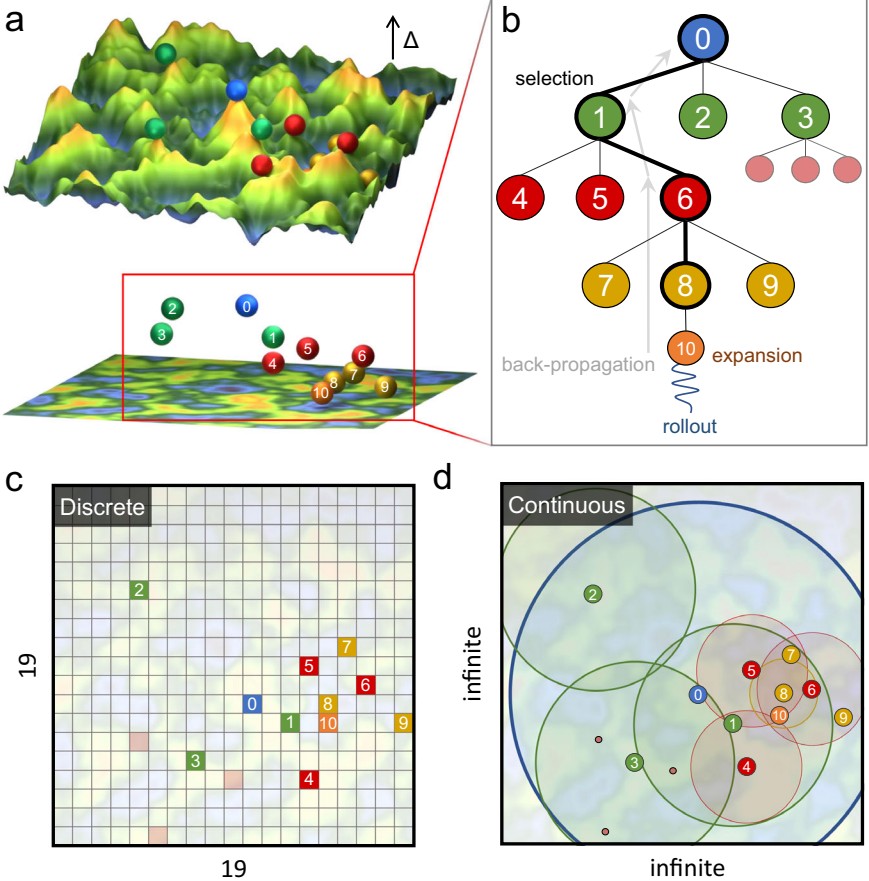

**Fig. 1 Schematic of the continuous action MCTS algorithm applied for exploration of high-dimensional potential parameter surfaces. a** Top: Simplistic representation of an objective landscape for a two-parameter search problem. In-plane axes correspond to (two) independent model parameters. The out-of-plane axis corresponds to objective values, which is defined as the weighted sum of the error in model predicted energies of clusters with respect to target energies. This objective is minimized by our c-MCTS algorithm. The spheres represent candidates of different model parameters within an MCTS run, where differences in their vertical positions indicate differences in their objective values. Bottom: Slightly tilted view of the above with the surface represented as a contour map below the spheres. The numbering on the spheres corresponds to their node positions in the MCTS tree shown in **b**. These numbers roughly correspond to the order that the candidates are explored. **b** Schematic showing the root, parent, child nodes, and their relationship within an MCTS tree structure. A typical MCTS search involves node selection, expansion, simulation (playout), and back-propagation. Different coloring of the nodes indicates different depths in the MCTS tree. The algorithm balances between exploration (lateral expansion of nodes) and exploitation (depth expansion of nodes). As shown in **a**, the objective value of an MCTS run is expected to decrease quickly along with the depth of the tree. **c** Search space of a traditional MCTS algorithm, e.g., game board, is discrete. In the context of parameter optimization, for two discrete parameters each of 19 possible values, the search space consists of a finite 361 search positions. **d** The problem of parameter search, such as the objective surface illustrated in **a**, generally involves parameters that are continuous, which corresponds to infinite possible search positions. We handle this challenge by applying a range-funneling technique to the MCTS algorithm where the search neighborhood at each tree-depth becomes smaller and smaller such that the algorithm can converge to the optimal solutions.

set of actions are possible irrespective of the current depth of the tree, thus, rendering the tree structure meaningless. Second, the search space of several physical problems displays inherent symmetries, which are not properly accounted for in the traditional MCTS. This leads to poor search efficiency, especially in high-dimensional problems, or sometimes unwanted convergence of different MCTS branches to the same solution. Lastly, performing random actions during the playout policy may be useful to learn expected returns for game playing, but do not translate well in high-dimensional continuous action space due to the well-known "curse of dimensionality"[21] (see Supplementary Methods).

We introduce three concepts to enable MCTS operations for continuous actions space, namely, the uniqueness function, a tree-depth-based adaptive action space, and a dimensionality-dependent playout policy. First, the uniqueness function $f(\theta_j)$ for a node $\theta_j$ returns a value between 0 and 1, with a higher value

indicating the uniqueness of this node with respect to all the other nodes in the tree. This not only helps avoid degenerate actions but also promotes explorations of those actions that are dissimilar to the previous ones, thereby efficiently sampling a larger search region. This is among the most essential feature of c-MCTS that enables high search efficiency. The definition of $f(\theta_j)$ used in this work (see Supplementary Methods) and its performance for high-dimensional continuous space is demonstrated in Supplementary Fig. 1 and Supplementary Table 1.

Second, an adaptive action space that depends on the tree-depth was included to provide better child-parent correlation within the MCTS tree structure. As illustrated in Fig. 1d, in this new scheme the range of possible actions from a node continually decreases as we go down the tree-depth. This ensures that the child node is based on actions that are within the search scope of its parent node. Further, this allows the MCTS algorithm to incrementally refine its search space; larger scans are made in the

initial phase of the search, followed by a more focused search in the identified interesting regions. This also restores a meaningful correlation between the parent and the child node and allows the algorithm to converge to a reasonable optimal solution as one moves down the tree. More details on the appropriate choice of the window scaling parameter with tree-depth is provided in the Supplementary Methods.

Third, we introduce an efficient MCTS playout policy for high-dimensional continuous action space. Traditional MCTS relies on a playout policy with random moves, but when the search space is intractable such a policy can make the learning process extremely difficult and inefficient[22,23]. For example, even with the tree-depth based scaling of action space, a random playout (assuming uniform probability distribution) will have a high chance of action selection which is far from the parent node; see Supplementary Methods (Section 3.2) for a discussion on how uniform or multivariate Gaussian probability distribution based playout policies systematically select actions that have large displacements from the parent node. To avoid this issue, we used a biased playout policy that selects action based on the probability distribution $r^{-(d-1)}$ where $r$ is the distance from the parent node, and $d$ is the dimensionality of the search space. Based on our experiments, we found this playout policy to provide significant improvement in the performance owing to better action selections.

We first demonstrate the efficacy of our approach by comparing its performance with other state-of-the-art evolutionary and other popular gradient-based approaches[24] on 25 well-known trial functions or artificial landscapes, containing deceitful local minima and a high-dimensional search space, (see Supplementary Fig. 1, Supplementary Table 1, and Supplementary Discussion (Section 4.1). Our c-MCTS algorithm outperforms other popular approaches used for materials applications both in terms of the solution quality (maximum reward) and search efficiency (low computational cost). Supplementary Discussion Section 4.2 highlights some of the known limitations of such methods, especially w.r.t scalability, problem dimensionality, and convergence. Especially for trial functions with higher dimensionality (e.g., >50), we note that the performance of c-MCTS is clearly superior to other representative and popular optimizers, illustrating the appeal of c-MCTS for multidimensional materials design and discovery problems. We also applied c-MCTS to classic reinforcement problems with a continuous action space (Supplementary Movies 1, 2, and 3) and demonstrate several other examples (please see Supplementary Discussion Section 4.3 Examples 1–4), where high-dimensional neural networks (NN) with up to a billion weights were trained successfully using c-MCTS.

**High-throughput navigation of high-dimensional PES**. As a representative impactful materials application, we next demonstrate the ability of our c-MCTS approach to successfully navigate through an 18-dimensional potential energy landscape (see Methods section and Supplementary Note 3—Section 7.1 for a description of the hybrid bond-order formalism) for 54 different elements that include alkali element, alkaline-earth elements, transition element, rare-earth, metalloids, and nonmetals. We assess the performance of the optimized hybrid bond-order potentials (HyBOP) (as tabulated in Supplementary Data 1) with respect to the ground truth, i.e., density functional theory (DFT) computed energies and forces of ~165,000 total configurations (including both low-energy near equilibrium (0.5–1.5 eV/atom range) and high-energy nonequilibrium configurations (1.0–12.0 eV/atom range) in our training (~95,000) and test (~70,000) datasets. As shown in Fig. 2a, b, ML optimized HyBOP

displays remarkable performance for all the elements across the periodic table. Mean Absolute Error (MAE) in cluster energies of Group IV-IB transition metals is ~90 meV/atom and ~88 meV/atom for the training and test set, respectively. Corresponding values for the post-transition metals from Group IIB (e.g., Zn, Cd) are slightly smaller at ~60 meV/atom and ~90 meV/atom, while elements in Group IVA (e.g., Pb) and Group VA (e.g., Bi) display MAE of ~65 meV/atom for both the training and test datasets. Nonmetals that include C, P, S, and Se show relatively higher average MAE of ~138 meV/atom on the training and test datasets, while metalloids such as B, Si, Ge, As, Sb, and Te, have an average MAE of ~100 meV/atom. Group IA alkali elements, including Li, Na, K, Rb, and Cs, have an average MAE of ~77 meV/atom on both the training and test datasets. Similarly, Group IIA elements from Be to Ba show an average MAE of ~51 meV/atom. Note that the MAE, as expected, are slightly higher for highly nonequilibrium clusters compared to the near-equilibrium configurations. Given that the DFT training data has a typical accuracy of 20 meV, our newly trained HyBOP models display remarkable performance in capturing vast regions of the energy landscape, with energies sampled from near-equilibrium to highly nonequilibrium cluster configurations and with sizes from dimers to bulk-like cluster configurations (>50 atoms).

We next assess the performance of our c-MCTS trained HyBOP models by comparing the forces experienced by atoms for the various sampled configurations with those computed from DFT. Note that forces were not included in the training. The high quality of our c-MCTS trained HyBOP models is evident from the strong correlation across a large number of none-quilibrium nanoclusters (total ~145,000 that have a wide range ~100 eV/Å for forces) in our test dataset (see Supplementary Note 5 (Section 9.2–9.55)). As seen in Fig. 2b, the trend in MAE prediction errors on forces is similar to that seen in energies. The Group IA alkali and Group IIA alkali earth metals such as Li, Na, K, Rb, Cs, Mg, Ca, Sr, and Ba have force MAE of ~187 meV/Å. Transition metals such as Cu, Ag, Au, Pd have marginally higher MAE ~300 meV/Å compared to the alkali and alkali earth elements. Magnetic elements such as Fe, Co, Mn, and Ni from transition block elements have an MAE ~450 meV/Å in force prediction. Metalloids (B, Si, Ge, As, Sb, and Te) from Group IIIA and Group VIA, respectively have an MAE ~800 meV/Å in force prediction (and MAE ~99 meV/atom in energy prediction). The block of transition metal elements containing 25 different elements with d electrons have an average MAE ~720 meV/atom in force predictions (MAE ~90 meV/atom in energies). Nonmetals such as C, B, P, S from Group IVA and Group VA display a strong correlation with DFT forces over a wide range (~200 eV/Å) but have a relatively higher MAE prediction error of 1300 meV/Å compared to the rest.

Notably, we find both the energy and force predictions of c-MCTS optimized HyBOP models to be significantly better than many other commonly available potential models for several of these elements. While not exhaustive, we compare the model performance for 38 different elements covering a range of existing potential types[25], such as EAM, MEAM, Tersoff, AIREBO, ADP, AGNI, and SW in Supplementary Fig. 2 and Supplementary Note 2 (Section 6.1–6.38). In comparison to the heat map shown in Fig. 2, Supplementary Fig. 2 depicts the corresponding MAE generated from some of these best-performing force fields available in the literature. We note that the errors in energies and forces predicted by the best existing potential are > > 220 meV/atom and > > 2252 meV/Å, respectively for the different elements shown in Supplementary Fig. 2. We also compare the errors obtained from existing high-quality state-of-the-art ML models (see Supplementary Figs. 3 and 4; details are provided in Supplementary Software 1). We find that even arguably the best

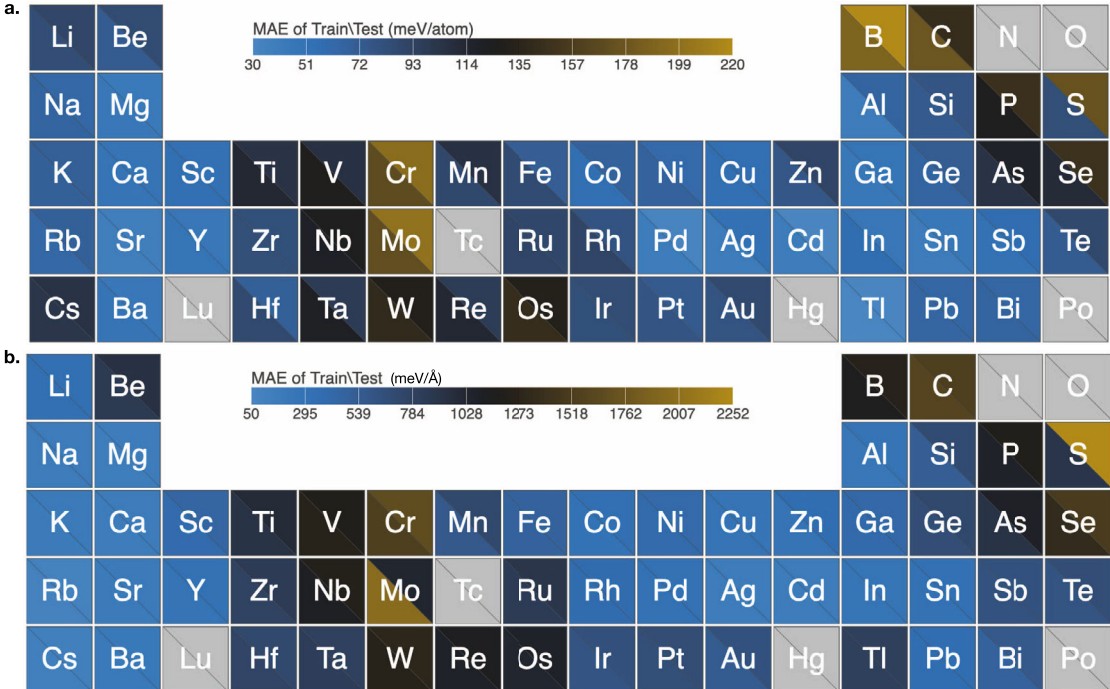

**Fig. 2 HyBOP model performance after high-throughput exploration of the potential energy surface of nanoclusters using continuous action MCTS.** A bond-order model with Tersoff-type formalism for 54 different elements across the periodic table is parameterized using an extensive training dataset generated from density functional theory (DFT) computations. The training set includes nanoclusters of varying sizes and shapes (~1500 configurations and their energies) for each of the different elements. Mean absolute error (MAE) in **a** energy and **b** force predictions of the bond-order force fields relative to the reference DFT model on a test set of ~70,000 configurations is shown using the color bar in meV/atom and meV/Å units, respectively.

available ML potential models such as GAP and SNAP have large errors (>1 eV/atom for energies and >1000 eV/Angstrom for forces) when representing the energies and forces of clusters far-from-equilibrium in our training and test set (please see Supplementary Figs. 3, 4 and Supplementary Note 1). This is much higher than those for our ML-trained models (see Fig. 2). This dramatic improvement in the performance can be attributed to two factors, first, the complex chemistry of the nanoclusters that is distinct from bulk-like behavior to which the existing potentials are primarily fit to, and second, the ability of the c-MCTS to find accurate fitting parameters as compared to human intuition. Further details on the MAE obtained for different systems using existing potentials are provided in Supplementary Note 2 (Section 6.1–6.38). In addition to the energies and forces, we also evaluate bulk properties (lattice constants[26], cohesive energies[26,27], and elastic constants[28–30]) predicted by our HyBOP model for ~200 different polymorphs of these 54 elements. We note that the inclusion of larger bulk-like cluster configurations (>50 atom clusters) in the training set and longer-range interactions in HyBOP ensured that bulk properties are reasonably captured by our models (please see Supplementary Figs. 5, 6 and Supplementary Note 6).

The methodology above can be easily extended to develop BOP models for alloy systems. As examples, we demonstrate the application to two binary alloys (i) one with solid solution (Ag-Au), another one that forms (ii) ordered structures (Al-Cu) over a wide composition range. These constitute a 54-dimensional parameter space. As seen from Supplementary Fig. 7, the performance of c-MCTS trained BOP models is quite exceptional across a broad composition range. To further test the scalability of c-MCTS, we have also trained the weights of high-dimensional neural networks with Behler–Parrinello symmetry functions for several different elements (Al, Mo, and C—please see Supplementary Note 8). We find that, even for high-dimensional NN with

1000's of weights, the performance of c-MCTS is comparable to state-of-the-art gradient-based high-quality optimizers such as ADAM[31]. It is however worth noting that the c-MCTS is a global gradient-free optimizer and is expected to work better for problems with ill-defined gradients compared to gradient-based approaches.

### Trends in elemental errors and their origins

We next aim to understand the relationship between the chemistry and the HyBOP model performance for any underlying element. We project the complete cluster test dataset in a 2D principal component (PC) space as shown in Fig. 3a. A popular fingerprinting scheme, SOAP or smooth overlap of atomic positions[32], that transforms the structural arrangement of a cluster to a unique vector representation was utilized (see Methods). A larger span in the PC space by an element is characteristic of its higher structural diversity; for example, some of the well-known elements with the most diverse chemical bonding, such as B, C, and S, can be seen to have a much larger PC span. We find a strong correlation of this PC area with the element position in the periodic table; moving down the columns of alkali, alkaline-earth, Group IIIA, Group IVA, Group VA, and Group VIA elements, we note that the configurational diversity systematically decreases. Similarly, moving across the 3d, 4d, and 5d transition rows, the spanned PC area first increases and then decreases in accordance with the number of valence d-electrons. A good match between the expected chemical diversity and the spanned PC area of an element is indicative of a well-sampled and comprehensive (training and test) cluster dataset. Another notable aspect from Fig. 3a is that many of the clusters with high prediction errors lie at the boundary of the PC space which are expected to belong to under-represented regions of the potential energy surface. An improvement in the model performance is thus expected if more clusters from such regions are included in the training dataset. When grouped according to the position in

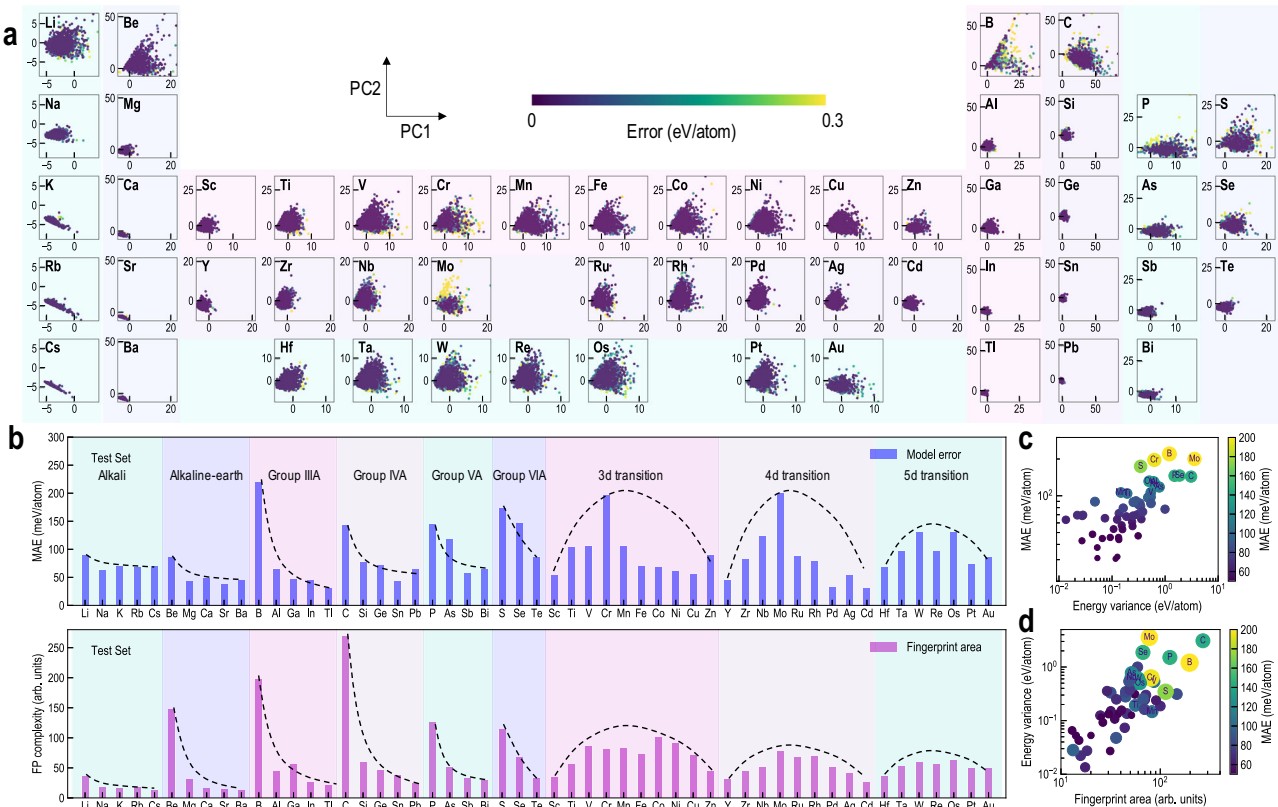

**Fig. 3 Trends in model prediction errors with structural diversity of the element. a** SOAP fingerprint representation of clusters across different elements as projected along with the first two principal components (PCs). The area covered by an element in the PC space is proportional to its structural diversity, and thus, the difficulty to train an accurate potential model. Also, the structural diversity of an element is closely related to its position in the periodic table. Regions of higher errors mostly occur at the boundary of the PC space which are expected to belong to under-represented configurations. **b** Trend in the prediction errors for different elements when grouped across different columns or rows of the periodic table. A similar trend can be seen for the elemental PC area, suggesting that elemental structural diversity correlates well with the model errors. The dotted lines capture the approximate trend within a group of elements and guide the eye. **c** Correlation between the prediction error and the inverse of the smoothness of the element energy surface. **d** Elements with high prediction errors display either high structural diversity or exhibit highly corrugated energy surfaces. Elements with mean absolute error (MAE) >100 meV/atom have been marked in **c** and **d**.

the periodic table, the errors in the model energy prediction roughly evolve as per the element chemistry (see Fig. 3b). Moving down the columns of alkali, alkaline-earth, Group IIIA, Group IVA, Group VA, and Group VIA elements the model prediction errors systematically decrease (within certain approximation), while moving across the 3d, 4d, and 5d transition rows, such errors first increase and then decrease. These trends are consistent with the above discussion on configurational diversity, as captured by the spanned PC area. To further quantify this correlation, the area in the 2D PC space was computed using a convex hull construction and can be seen to match well with the errors in model predictions (Fig. 3b)—with the notable exception of Mo, which had a relatively high model error than expected from the respective configurational diversity value. We argue that developing potentials for elements that show large configurational diversity, or large PC area, should be more difficult. This is because finding a unique set of the HyBOP parameters that capture all such high-energy regions of the energy surface is nontrivial.

Another aspect that can make potential learning more difficult is the smoothness of the energy surface. This implies that small perturbations in an element's configuration leads to energy changes with high variance. Developing potential for such a system would be difficult as the underlying energy surface would be highly corrugated with strong variations in energy depending on the direction of the perturbation, or could contain multiple local minima. Based on the principle of a variogram[33], the variance in energy changes for an element was computed by measuring the corresponding SOAP fingerprint distances between different, but similar, configurations. As shown in Fig. 3c, elements with large prediction errors also displayed high variance in the energy changes and vice versa. Combining both the above aspects in Fig. 3d, it can be seen that models with high prediction errors belong to cases with either high structural diversity or highly corrugated energy surface; almost all elements with MAE >100 meV/atom lie in the upper right corner of Fig. 3d. Therefore, this analysis not only explains the observed trends in the model prediction errors across the periodic table but also provides confidence that the developed c-MCTS search successfully found highly optimal HyBOP parameters in all cases.

**Dynamic stability of clusters to bulk, relative isomer stabilities, normal mode analysis.** We perform a rigorous test of our c-MCTS-optimized HyBOP potentials by evaluating the dynamic stability of the 54 elemental nanoclusters with different topologies and over a broad size range (5 to 50 atoms) at different temperatures (see Supplementary Table 9). We perform MD simulations and analyze the dynamical stability for most clusters using the mean square deviation (MSD) of the atoms during a 1 ns

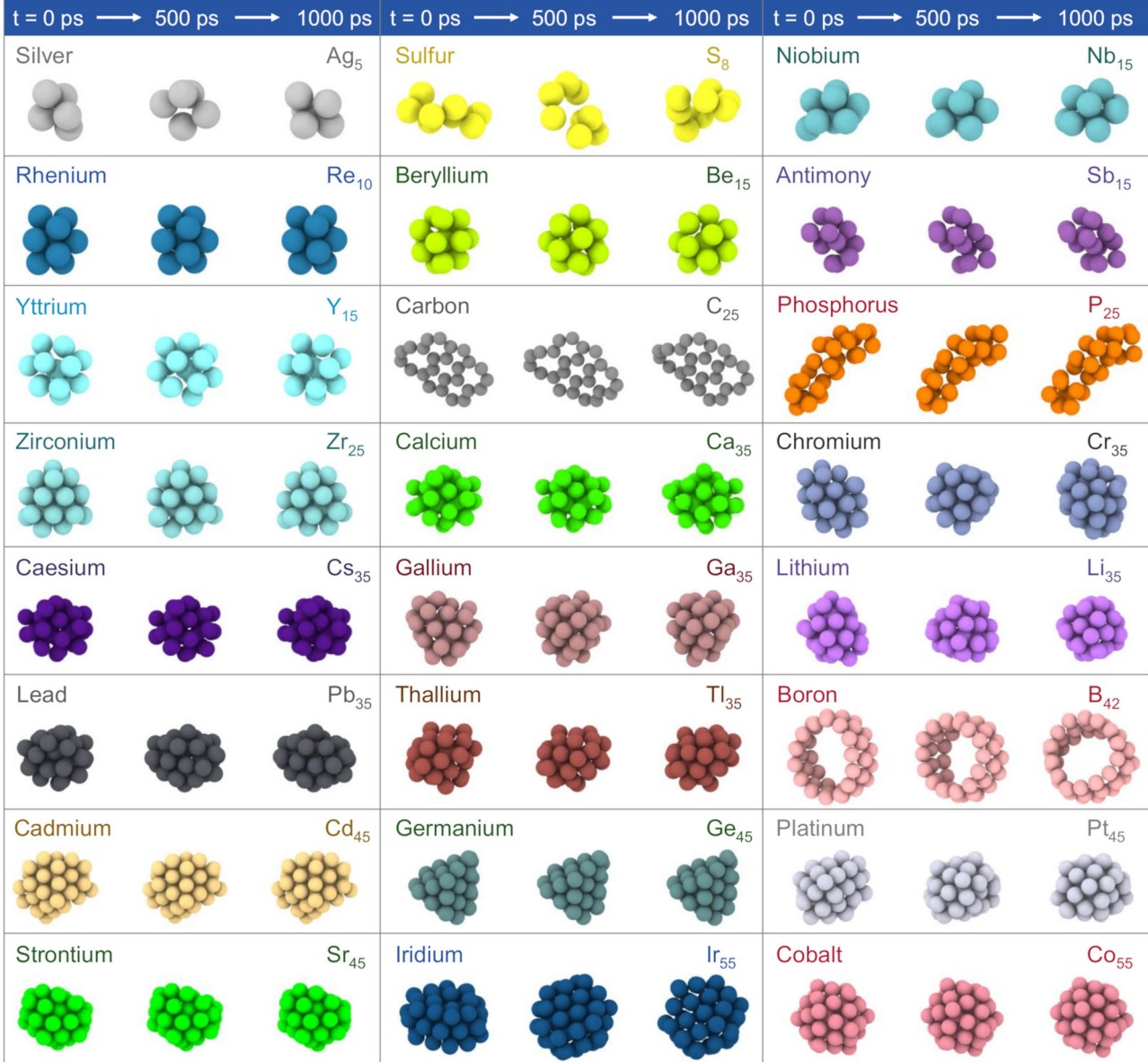

**Fig. 4 Dynamic stability of the clusters of various representative elemental systems.** Snapshots sampled from 1 nanosecond MD simulation trajectories illustrate the dynamic stability of the clusters of representative topologies and sizes for the different elements. To rigorously test the dynamic stability, MD simulations were performed for >40,000 nanoclusters for the 54 different elements in different size ranges. More detailed MD trajectories for each element and the simulated temperatures are included in the Supplementary Note 7 Section 11.3.

trajectory. A cluster is considered to be dynamically unstable if the MSD is greater than 2 Å; for a few special configurations, such as rings, visual inspection of energy or configuration trajectory was used to evaluate dynamical stability as system rotations lead to artificially higher values of MSD. We find that the c-MCTS optimized models capture the dynamical stability across >40,000 clusters tested, with a few representative trajectories shown in Fig. 4. Detailed trajectories for each element are included in Supplementary Note 7 (Section 11.3). This rigorous validation provides confidence for the future use of the developed potential models as well as the c-MCTS framework for exploring various structural and dynamical properties of nanoscale systems. We further perform relative stability analysis for representative isomers of elemental clusters from each group and compare their relative energy ordering from the HyBOP model with that from DFT (see Methods and Supplementary Note 7 for details). As seen from Supplementary Fig. 9 and Supplementary Note 7

(Section 11.4), the agreement is quite remarkable given the subtle differences that exist in many of these isomers. Finally, we also perform normal mode analysis for several clusters of representative elements. We observe that the computed normal modes from HyBOP match very well with the DFT computed normal modes (see Supplementary Fig. 8 and Supplementary Note 7 (Section 11.1–11.2)) over a broad frequency range, highlighting the high quality of our HyBOP models.

In summary, we build on the powerful ideas of reinforcement learning and decision trees to develop an effective search algorithm in high-dimensional continuous action space (c-MCTS). Our algorithm extends Monte Carlo tree search to continuous action spaces with three novel concepts (uniqueness criteria, window scaling, and adaptive sampling) to accelerate the search. c-MCTS broadly outperforms state-of-the-art meta-heuristic and other optimization methods. We applied this method to develop accurate bond-order potentials (with 18-

dimensional search space) for 54 elements across the periodic table, a rather nontrivial feat which would have taken years of effort with the traditional approach. On the one hand, the developed potentials will be of use to the materials simulation community owing to their accuracy in capturing the energy and atomic forces across large configuration space, making them attractive in the field of catalysis, especially for problems involving single-atom-catalysts[34,35] that form localized active sites. On the other hand, the developed c-MCTS will be beneficial in solving grand challenges in materials discovery that often involve search in a continuous space.

## Methods

**Generation of diverse cluster configurations.** The accuracy and transferability of a potential model or force field depends on the quality of training and test datasets. In this study, our training and test dataset consists of ~95,000 and ~70,000 clusters configurations and their energies for 54 different elements across the periodic table, respectively. The sampled configurations contain (a) the ground state atomic structures (b) atomic structures near equilibrium, and (c) structures far-from-equilibrium. A total of ~19,000 ground state cluster configurations were evenly included amongst the training and test dataset. As a result, the DFT computed energies of sampled configuration generates a broad spectrum of energy window (ranging from ~1 eV/atom, for alkali and alkali earth metals to ~14 eV/atom, for nonmetals) in the training and test dataset. A continuous energy window from low to high for the sampled configurations ensures that the energetics and dynamics of clusters configurations starting from equilibrium to far-from-equilibrium are adequately represented. Ground state clusters configurations are mined from different literature[36] as well as from our own calculations using a standard atomic structure prediction method (such as genetic algorithm[37,38]). We used Boltzmann-based Metropolis sampling and a nested ensemble-based[39–41] approach to generate the cluster configurations that have energies near equilibrium and far-from-equilibrium. The combination of these two techniques generates cluster configurations of a continuous range of energies from high to low. From the size perspective, the cluster configurations sampled consist of dimers to bulk-like configurations i.e., clusters containing atoms more than 50 atoms. Details on the various sampling approaches to generate cluster configurations are provided in Section 4.4 of Supplementary Discussion.

**Reference energy and force computations for cluster configurations.** The energies of all the cluster configurations were computed using density functional theory (DFT) as implemented in the *Vienna Ab initio Simulation Package* (VASP)[42]. The low-energy cluster configurations are further relaxed using conjugate gradient approach[43] whereas in the case of nonequilibrium clusters we perform static DFT calculation to compute their energies and forces. To avoid interactions between two periodic images we assign the box length such that the distance between two periodic images is at least 15 Å or larger. The DFT calculations were performed using the generalized gradient approximation (GGA) with the Perdew–Burke–Ernzerhof (PBE) exchange-correlation functional, and the projected-augmented wave (PAW) pseudo-potentials[44]. The details of pseudo-potentials used in our calculations are provided in Supplementary Table 2 in Supplementary Discussion Section 4.5. Spin-polarized calculations with the Brillouin zone sampled only at the Γ-point were performed. To handle errors that may arise during the structural relaxation or static DFT calculations, our high-throughput workflow used an in-house python wrapper[45] around VASP along with a robust set of error handling tools.

**Learning of trial functions.** We performed a rigorous test using a series of trial functions to demonstrate the suitability of our c-MCTS approach. We used several popular test functions[46] which were designed to represent a wide variety of different continuous action surfaces. These surfaces include many local minima (Ackley, Buckley, Ragstrigin, etc.), misleading or nonexistent gradients (Hyper Plane, Dejong Step Function), or other functions that are designed to mimic common optimization traps that one may encounter. An advantage of these functions is that they are computationally quick to evaluate and have known solutions. In addition, several of them can be scaled up to an arbitrary number of dimensions. As such, these trial functions provide a series of comprehensive first-order tests to evaluate a search algorithm. These functions have implementations in a wide variety of programming languages such as MATLAB, Python, R to name a few. We compare the performance of c-MCTS with several other global/local optimizers including evolutionary algorithms such as particle swarm, Bayesian optimization, and random sampling. For the particle swarm, the PySwarm optimization package[47] was used with 50 particles per-trial and the hyperparameters (c1, c2, w) were adjusted on a per-trial basis starting from the default of (0.5, 0.3, 0.9). For the Bayesian-based optimization the HyperOpt Python package[48] with the TPE. Suggest option was used as the trial selection method. The performance of the various optimizers on several different trial functions is listed in Supplementary

Table 1 and the convergence to solution for representative cases is shown in Supplementary Fig. 1.

**Model selection—Hybrid bond-order potential.** We navigate an 18-dimensional potential energy surface represented by a hybrid bond-order potential (HyBOP) expressed as

$$V = V_{SR} + V_{LR} \qquad (2)$$

which utilizes a Tersoff formalism[49] for describing short-range directional interaction ($V_{SR}$) and a scaled Lennard–Jones (LJ) function for long-range interaction ($V_{LR}$). The short-range pair potential function $V_{pair}$ is described by

$$V_{SR} = \frac{1}{2}\sum_i\sum_{i\neq j}f_C(r_{ij})[f_R(r_{ij}) + b_{ij}f_A(r_{ij})] \qquad (3)$$

where $f_C(r_{ij})$, $f_R(r_{ij})$, and $f_A(r_{ij})$ are the cutoff, repulsive, and attractive pair interactions, respectively, between atoms $i$ and $j$ separated by a distance $r_{ij}$, and $b_{ij}$ is a bond-order parameter which modifies the pair interaction strength between atom $i$ and $j$ depending on their local chemical environment. The long-range interactions $V_{LR}$ uses a scaled Lennard–Jones (LJ) function which is given by,

$$V_{LR} = \sum_i\sum_{j>i}4\epsilon_{ij}f_s(M_i)\left[\left(\frac{\sigma_{ij}}{r_{ij}}\right)^{12} - \left(\frac{\sigma_{ij}}{r_{ij}}\right)^6\right] \qquad (4)$$

where, $\epsilon_{ij}$ and $\sigma_{ij}$ are LJ parameters for a pair of atoms $i$ and $j$ that are a distance $r_{ij}$ apart. $f_s(M_i)$ is a scaling function that describes the dependence of LR contribution from a given atomic pair $i - j$ on the number of atoms within a prescribed radial distance $R_c^{LR}$ from the atom $i$. Details of the potential model are provided in Supplementary Note 3 (Section 7.1). Owing to its flexibility in describing various bonding characteristics as well as to facilitate comparisons, we retain the HyBOP formalism across all the elements in the periodic table.

**Learning of high-dimensional potential energy surface (PES).** To navigate the high-dimensional continuous parameter space, we developed a workflow that interfaces the MCTS algorithm with a molecular simulator, a large-scale atomic/molecular massively parallel simulator (LAMMPS)[50]. While MCTS is used to navigate the high-dimensional HyBOP parameters space, LAMMPS is used to evaluate the performance of a given input set of HyBOP parameters by computing the mean absolute error (MAE) of energies for the clusters in the training dataset. Each node of the MCTS search represents a particular set of HyBOP parameters, with the reward $r$ in Eq. (1) evaluated as the weighted sum of MAE error in the cluster energies as discussed in Supplementary Note 4 (Section 8.1–8.2). MCTS is a decision tree-based approach (comprising of selection, expansion, simulation, and back-propagation) that builds a shallow tree of nodes where each node represents a point in the search space and downstream pathways are generated by a playout procedure. The algorithm simultaneously explores potentially better pathways to reach the optimal point in a search space and exploits a subset of pathways that have the greatest estimate values of the search function. This combination of exploration vs exploitation and an appropriate trade-off mechanism between them are found to be the most efficient strategy of identifying optimal points for a given function. This strategy is also extended to optimize weights of high-dimensional neural networks. The details of the c-MCTS approach to learn high-dimensional PES are provided in the Supplementary Methods.

**Fingerprinting and principal component analysis.** To analyze the structural diversity of the cluster dataset across the different elements, we utilized the smooth overlap of atomic positions (SOAP) fingerprinting method[32] as implemented in the DScribe python library[51]. SOAP encodes the atomic neighborhood around a spatial point (or an atom) using the local expansion of a Gaussian smeared atomic density with orthonormal basis functions composed of spherical harmonics and radial basis functions. The advantage of using the SOAP fingerprint is that it is invariant to translation, rotation, and permutations of alike atoms of a configuration, and forms the basis for the development of several successful ML-based inter-atomic potentials. The following parameter settings were used for the SOAP fingerprinting: rcut = 6 Å, nmax = 6, and lmax = 4, where rcut is the cut-off radius for the atomic neighborhood around the concerned atom, namx is the number of radial basis functions (spherical Gaussian type orbitals) and lmax is the maximum degree of spherical harmonics. This resulted in a SOAP fingerprint vector for each atom, which was averaged using the "inner" averaging scheme (average over atomic sites before summing up the magnetic quantum numbers) to obtain a 105-dimensional configuration fingerprint for each cluster. Principal component analysis (PCA) was performed using the (standard) normalized SOAP fingerprint representation of the entire cluster dataset. The variance in energy changes as a function of SOAP fingerprint distances (cosine) was computed using the open-source Scikit-GStat Python library[33].

## Data availability
The data that support the findings of this study are available from the authors upon reasonable request.

## Code availability

The c-MCTS algorithm is implemented in the BLAST framework developed by the authors. The codes, scripts, and framework are available from the authors upon reasonable request.

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

## Acknowledgements

This material is based upon work supported by the US Department of Energy, Office of Science, Office of Basic Energy Sciences Data, Artificial Intelligence, and Machine Learning at DOE Scientific User Facilities program under Award Number 34532. This work was performed in part at the Center for Nanoscale Materials, the Advanced Photon Source, and the Center for Nanophase Materials Sciences, which are US Department of Energy Office of Science User facilities supported by the US Department of Energy, Office of Science, Office of Basic Energy Sciences, under Contract No.s DE-AC02-06CH11357, DE-AC02-06CH11357, and DE-AC05-00OR22725, respectively. This research used resources of the National Energy Research Scientific Computing Center, which was supported by the Office of Science of the US Department of Energy under Contract No. DE-AC02-05CH11231. An award of computer time was provided by the Innovative and Novel Computational Impact on Theory and Experiment (INCITE) program of the Argonne Leadership Computing Facility at the Argonne National Laboratory, which was supported by the Office of Science of the US Department of Energy under Contract No. DE-AC02-06CH11357. SKRS would also like to acknowledge the support from the UIC faculty start-up fund. This work was supported by the United State Department of Energy through BES award DE-SC0021201. This research used resources of the National Energy Research Scientific Computing Center (NERSC), a US Department of Energy Office of Science User Facility located at Lawrence Berkeley National Laboratory, operated under Contract No. DE-AC02-05CH11231. The authors thank Dr. Badri Narayanan (University of Louisville) for useful discussions. The authors

thank Prof. Jörg Behler (Georg-August Universität Göttingen) for providing NN program using RuNNer.

## Author contributions

S.M., R.B., and S.S. conceived and designed the project. T.D.L. and R.B. contributed equally. T.D.L. developed the RL framework for continuous action space problems with input from S.M., H.C., and S.S. T.D.L. evaluated the performance of the high-dimensional trial functions. S.M. deployed the RL c-MCTS algorithm and trained the bond-order potential functions for the various elements across the periodic table. All the first-principles datasets were generated by S.M. Validation of the force fields were carried out by S.M. and S.B. R.B. developed an ML framework to carry out the analysis of the error trends in the potential functions across the different elemental systems. S.B. performed high-throughput simulations to evaluate the dynamical stability of all the different elemental clusters. H.C. integrated the c-MCTS algorithm into the BLAST multi-fidelity framework. T.D.L. carried out validation of the c-MCTS on training high-dimensional neural networks. All the authors contributed to the data analysis and to the preparation of the manuscript. S.M., R.B., and S.S. wrote the manuscript with input from all the coauthors. B.V., K.S., M.S., T.P., M.J.C., S.K.G., and B.G.S. provided feedback on the manuscript. S.S. supervised and directed the overall project.

## Competing interests

The authors declare no competing interests.
