## [Peer Review File · Nature Communications]

Learning in continuous action space for developing high dimensional potential energy modelsEditorial Note: This manuscript has been previously reviewed at another journal that is not operating a transparent peer review scheme. This document only contains reviewer comments and rebuttal letters for versions considered at *Nature Communications*. Mentions of prior referee reports have been redacted.

REVIEWERS' COMMENTS

Reviewer #1 (Remarks to the Author):

The authors have worked hard to address the issues I raised - their responses are extensive and convincing. I recommend this paper be accepted.

Reviewer #2 (Remarks to the Author):

The authors have replied at length to the points raised at the previous **[REDACTED]** review. I was positive to the science, but gave a suite of comments.

Many replies are very useful, but not all.

They have included new data in short time on "binary alloys" and some NN computations.

I guess, these showcase further the robustness of the methodology.

And they highlight the accuracy improvements relative to previous work.

So the science is even better.

However, for most part I don't have the patience to continue in detail on the debate, and I don't think it is my role to iteratively improve their work until it is acceptable.

So I stick to a few important points:

I am still 100% firm on that they cannot use the title

"...Determination of High Dimensional Potential Energy Surfaces".

For chemistry oriented readers this suggests way!!! too much.

They may call it something else, like atomic and binary potentials or whatever.

I acknowledge that their binary alloys and NN are very good additions - but it is so far from solving and validating that their approaches solves

the problem for general High Dimensional Potential Energy Surfaces(PESs).

Because they are still learning simple functions that represent the surfaces in the "force field" sense, and far from showing that they can learn a general complicated molecular PES accurately.

Related to this:

I think the accuracy discussion is improved - but not in any form I trust ALONE to be generalized to give general accurate "High Dimensional Potential Energy Surfaces" - notice accuracy is not improved, only the discussion of it and more contents added.

I am still not completely clear on why this research could not be better represented in two papers in more specialised journals where equations and arguments could take more space for the method part, and more space given to the potential part.

Fundamentally: I think one should have intelligible contents in journal articles, and not in the discussion during review process and even to limited extent in supplementary information.

OK, I have reached a point where the case is about even for me wrt this paper:

It is significantly improved, and could be published. I think the science

is actually great, but there is a certain "put a lot of stuff in very little space"

hype to it, that I anticipate is related to the author tries to get accepted in a high profile journal.

But really, that is not necessary the optimal way ahead for progress of science.

To be acceptable for publication they must fix the tone

of "we can do any PES very accurately", including the title.
I could then live with the short and low level discussion of methods.
But, I will safely leave to the editor whether it should
be best published in two specialised papers or one in nature communications.

Reviewer #3 (Remarks to the Author):

The authors have addressed most of the issues that I raised. At this stage, although I feel that in many places the new numerical tests are still unfair and that the combination of the methodology and the application indeed weakens the paper from both perspectives, my feeling may be subjective. As such, I would ultimately suggest publishing the paper on Nature Communications.

One last thing that I think would be extremely important is that the authors should publish their data and necessary testing tools to reproduce all their results. They compared their results with both deep learning methods like Adam and MLIP methods like GAP, but things like parameter setups may not be optimal. Since these results, together with the authors' own results are very important for the validity of the paper, making these things publicly available would be necessary. This will also help improve the transparency and reproducibility of the work.

1 Author Rebuttals to Reviewer's Comments:

We sincerely thank the reviewer for their affirmation and recognition of the quality of this work. We have responded in detail to each of the referee's comments below, where the referee comments are in black font labeled Comment #.#, and our responses are in blue font.

Reviewer 1

Comment 1.1 : The authors have worked hard to address the issues I raised - their responses are extensive and convincing. I recommend this paper be accepted.

Reply: We thank the Reviewer for his/her kind words regarding our manuscript, and for accepting it in its current form.

Reviewer 2

Comment 2.1 : The authors have replied at length to the points raised at the previous [REDACTED] review. I was positive to the science, but gave a suite of comments. Many replies are very useful, but not all. They have included new data in short time on "binary alloys" and some NN computations. I guess, these showcase further the robustness of the methodology. And they highlight the accuracy improvements relative to previous work. So the science is even better. However, for most part I dont have the patience to continue in detail on the debate, and I don't think it is my role to iteratively improve their work until it is acceptable. So I stick to a a few important points: I am still 100% firm on that they cannot use the title "...Determination of High Dimensional Potential Energy Surfaces". For chemistry oriented readers this suggests way!!! too much. They may call it something else, like atomic and binary potentials or whatever.

Reply: We thank the referee for his/her positive assessments of our work. We have changed the title to "Learning in Continuous Action Space for Development of High Dimensional Potential Energy Models".

Comment 2.2 : I acknowledge that their binary alloys and NN are very good additions - but it is so far from solving and validating that their approaches solves the problem for general High Dimensional Potential Energy Surfaces(PESs). Because they are still learning simple functions that represent the surfaces in the "force field" sense, and far from showing that they can learn a general complicated molecular PES accurately. Related to this: I think the accuracy discussion is improved - but not in any form I trust ALONE to be generalized to give general accurate "High Dimensional Potential Energy Surfaces" - notice accuracy is not improved, only the discussion of it and more contents added.

Reply: We believe that the approach we have presented is scalable to high dimensionality and can learn complex PES for molecules as well. However, we agree that we have not demonstrated this since our focus was on systems used in materials science applications. We have rigorously demonstrated our approach to learn robust bond-order potentials for 54 elements, 2 binary alloys, as well as high dimensional NN for few

representative elements with the accuracy being better/at par with some of the best performing physics-based or data-driven potentials (MLIP) available. As per the reviewer suggestion, we have changed the title to focus more on the "potential model development" rather than "potential energy surface".

Comment 2.3 : I am still not completely clear on why this research could not be better represented in two papers in more specialised journals where equations and arguments could take more space for the method part, and more space given to the potential part. Fundamentally: I think one should have intelligible contents in journal articles, and not in the discussion during review process and even to limited extent in supplementary information. OK, I have reached a point where the case is about even for me wrt this paper: It is significantly improved, and could be published. I think the science is actually great, but there is a certain "put a lot of stuff in very little space" hype to it, that I anticipate is related to the author tries to get accepted in a high profile journal. But really, that is not necessary the optimal way ahead for progress of science. To be acceptable for publication they must fix the tone of "we can do any PES very accurately", including the title. I could then live with the short and low level discussion of methods. But, I will safely leave to the editor whether it should be best published in two specialised papers or one in nature communications.

Reply: We thank the Referee for their assessment and comment. We believe that both the approach and the application would have a broader impact on the materials design as well as molecular simulations community and as such would appeal to a wide Nat Comm readership across several disciplines. Specifically, we note that: i) A new RL learning framework (c-MCTS) is introduced to allow autonomous decision making and searching in continuous action spaces which is of central importance to many areas of physical sciences and real world problems, (ii) we show how the c-MCTS approach can be used to navigate high-dimensional nanoscale energy surface and construct high-quality interatomic potentials (from bond order to NN) for dozens of different elemental systems, and (iii) we highlight the trends and origin of errors in the trained models across the various elements in the periodic Table.

Reviewer 3

Comment 3.1 : The authors have addressed most of the issues that I raised. At this stage, although I feel that in many places the new numerical tests are still unfair and that the combination of the methodology and the application indeed weakens the paper from both perspectives, my feeling may be subjective. As such, I would ultimately suggest publishing the paper on Nature Communications.

Reply: We thank the Reviewer for their comments, and for recommending acceptance.

Comment 3.2 : One last thing that I think would be extremely important is that the authors should publish their data and necessary testing tools to reproduce all their results. They compared their results with both deep learning methods like Adam and MLIP methods like GAP, but things like parameter setups may not be optimal. Since these results, together with the authors' own results are very important for the validity of the paper, making these things publicly available would be necessary. This will also help improve the transparency and reproducibility of the work.

Reply: As per the reviewer suggestion, the NN potentials developed in this study and the other available MLIP methods such as GAP, SNAP, qSNAP, and MEGNET used for benchmarking are now provided in the "Supplementary Software" section. The reinforcement learning search for force field parameters is implemented in the BLAST software framework and is available from the authors via user proposal system at the Center for Nanoscale Materials at Argonne National Laboratory - a DOE scientific user facility. All the data that support the findings of this study are available from the authors upon reasonable request.